# In-context Learning for Zero-shot Medical Report Generation

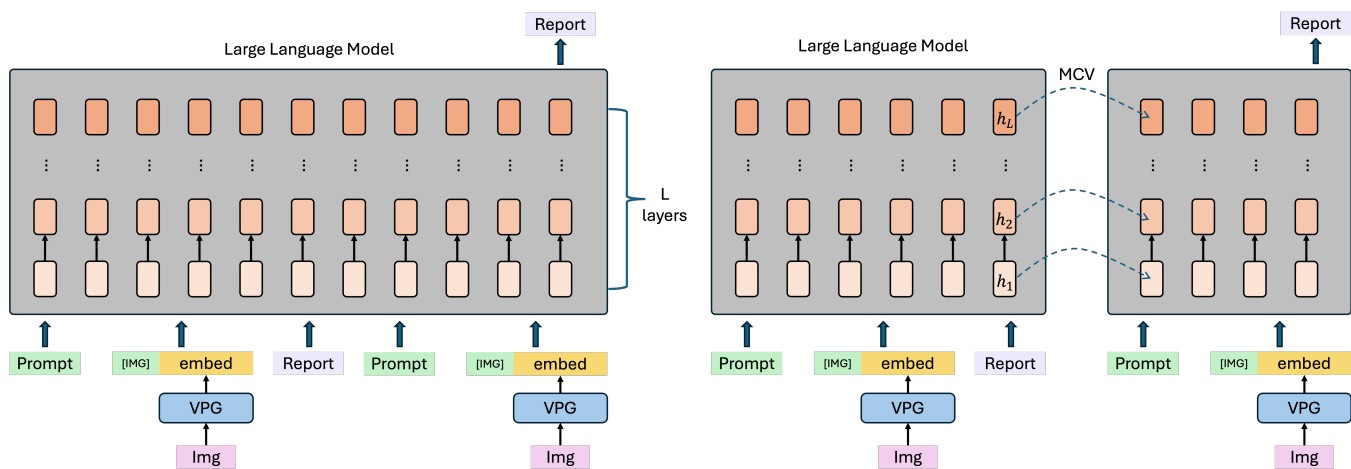

Figure 1: Comparison of multi-modal in-context learning and our proposed MCV. (a) The architecture of adopting multi-modal in-context learning for MRG. (b) The paradigm for our proposed MCV to alter multi-modal in-context learning. We attain the MCV from the last hidden states of a forward pass on the multi-modal demonstrations. One demonstration consists of a prompt, an example image and the corresponding report. *Img* refers to the image features encoded by vision encoders, [IMG] is a specific image proxy for LLMs to differentiate between visual and text embeddings.

## ABSTRACT

Medical report generation (MRG) has emerged as a pivotal research topic in the medical multi-modal field, given its potential to alleviate the heavy workloads of radiologists. Recently, advancements have been made with MRG systems that leverage large multimodal models (LMMs) to generate high-quality reports. To address the challenge of collecting large amounts of paired medical image-report data for training, this paper proposes a zero-shot report generation model based on in-context learning, we call it MCVGen. Departing from traditional in-context learning approaches that directly feed all demonstrations to a pre-trained large model, this work innovates by employing a multi-modal contextual vector (MCV) to represent the contextual information of demonstrations. Initially, we pre-train a medical large multi-modal model (Med-LMM) and secure the last hidden state of each demonstration through the forward pass in Med-LMM. Benefits from the auto-regressive mechanism, the last hidden state garners critical information to the targeted scenarios. Subsequently, we average the multiple MCVs and integrate them with the first hidden state on the new query, thereby shifting the latent states and guiding the model toward acquiring previously unlearned multi-modal contextual information. This approach has the advantage of regulating the number of prompts, thus reducing computational costs. We tested our model on the publicly available Open-IU and MIMIC datasets, demonstrating its exceptional zero-shot capability on both cross-center and cross-disease evaluations. We hope it could be a viable solution for practical clinical applications.

## CCS CONCEPTS

• **Computing methodologies → Computer vision**.

## KEYWORDS

Multi-modal in-context learning, Zero-shot, Medical report generation, Large multi-modal model

## 1 INTRODUCTION

Medical report generation (MRG) [31] stands as a core research topic within the medical multi-modal field, aiming to empower models to automatically generate reports that articulate findings from medical examinations. This endeavor seeks to lessen the heavy workload shouldered by radiologists.

Deep learning-based MRG models predominantly adopt an encoder-decoder architecture, where the capacity of the decoder to process lengthy sequences is paramount due to the typically extensive nature of medical reports. Initial efforts saw the deployment of

LSTM [10] networks, including two-level LSTM configurations [16, 19], as decoders for generating reports. However, the advent of the Transformer [38] marked a pivotal shift in decoder technology preference, leading to the development of variants like memory-driven Transformers [4, 5, 33], knowledge-enhanced Transformers [12, 23, 27], and those pre-trained on contrastive learning paradigms [22, 25, 26]. More recently, with large language models (LLMs) demonstrating remarkable linguistic processing capabilities, an increasing number of studies have begun utilizing these large models as decoders. By ingeniously crafting prompts infused with prior knowledge [3, 15] and applying fine-tuning [41] techniques, these novel approaches have proven effective in producing high-quality medical reports, showcasing the increasing reliance on advanced LLMs. However, collecting such large amounts of paired medical image-report data to support the fine-tuning of these models is expensive and challenging. For example, smaller centers may not be able to gather a sufficient number of paired samples. Additionally, for some novel diseases, the number of samples available in clinical settings is inherently limited. Therefore, developing MRG models with zero-shot learning capabilities holds greater potential for clinical applications.

Existing work proposed to reduce reliance on paired image-text data can be categorized into two categories. The first one utilizes knowledge graph [14, 28] to bridge visual and textual data. For instance, Liu *et al.* [28] proposed an auto-encoding knowledge graph for encoding reports or images, enabling the training of report generation capabilities by merely reconstructing the encoded knowledge graph back to text. The second type replaces the golden-labeled reports with more readily available supervised signals. For example, Han *et al.* [8] trained a model to first detect and segment spinal regions, then convert the detected and segmented information into reports. The primary reason these models require additional information is that the adopted decoders must undergo training to acquire the ability to decode visual features and generate reports. In this paper, we adopt a more direct approach by leveraging the exceptional zero-shot capabilities [43] of LLMs to design a zero-shot MRG model.

In-context learning [7] represents a striking property for LLMs to harness their zero-shot capabilities. It involves adapting a pre-trained model to new tasks by providing a prompt and some demonstrations without updating its parameters or adding extra parameters like LoRA [11]. However, directly applying in-context learning to MRG tasks encounters two main challenges. First, traditional in-context learning is not efficient. Medical reports are typically lengthy, and when combined with prompts and corresponding image tokens, the length of a single demonstration can range between 450 to 700 tokens. This significantly increases the computational burden and requires models with larger parameters to process such long sequences. Second, the effectiveness of in-context learning is unstable, heavily relying on the selection of templates and examples. Especially adopting the in-context learning in multi-modal domains [44], shuffling the order of demonstrations can also affect predictions.

To address these challenges, this paper introduces a novel approach named MCVGen, which leverages the zero-shot capabilities of LLMs through the MCV to generate medical reports. The objective of the MCV is to represent essential multi-modal contextual

information from a given demonstration and provide this information to guide new query report generation in a controllable manner. Similar concepts, such as *e.g.* task vectors [9], function vectors [36] and in-context vectors [29], have been employed in natural language processing (NLP) tasks. To apply this concept to MRG tasks, our methodology initially employs a forward pass on each demonstration, comprising instructions, example image, and corresponding report, through a LLM to create the MCV from the last hidden states (see Fig.1). In order to refine control over the generation process, the MCV preserves all the hidden states from each large language model layer, thus serving as vector represented instructions for the new query. For multiple demonstrations, the mean of all MCVs is computed to compile comprehensive intended task information. Subsequently, the resultant MCV is seamlessly integrated with hidden states on the first query token position, thereby initiating a pivotal shift in the generation process. Our proposed MCVGen offers three main advantages: 1) It does not entail extra computational cost, achieved without the concomitant elongation of processed sequence length. 2) Our approach enhances efficacy in controlling the generation process by embedding MCV information within each hidden layer. 3) MCVGen ensures that the new query outputs remain unaffected by the order and templates of the demonstrations.

We conducted experiments on the publicly available IU-Xray [6] and MIMIC [17] benchmarks and propose two zero-shot settings, namely cross-center and cross-disease, to validate the model's zero-shot capabilities. For the cross-center experiments, we first trained a Med-LMM model capable of report generation using data from one center and then tested it on data from another center. For the cross-disease tests, we utilized CheXpert [13] to label diseases and employed a leave-one-out approach, selecting 13 disease types for training and the remaining disease type for testing in each iteration. The experimental results demonstrate that our MCVGen exhibits exceptional zero-shot MRG capabilities. It can generate reports with different writing styles or describe novel diseases based on a few demonstrations, achieving state-of-the-art (SOTA) zero-shot MRG performances and comparable performance compared to existing fully supervised MRG systems.

Our contributions can be summarized as follows:

- To the best of our knowledge, we are the first work to leverage the LLMs for zero-shot medical report generation.
- We propose MCVGen, which introduces the MCV to alter multi-modal in-context learning for zero-shot MRG, making it efficient, controllable, and robust.
- We evaluate our proposed MCVGen on two zero-shot settings. MCVGen achieves SOTA zero-shot performances that are even comparable to some fully supervised MRG systems.

## 2 RELATED WORK

### 2.1 Alternative Learning Paradigms for Medical Report Generation

Existing SOTA MRG systems can generate logically coherent and factually correct reports, benefiting from several successful concepts, such as memory-driven modules [4, 5, 33], contrastive learning [22, 25, 26], fine-grained lesion features [21, 35], and medical knowledge injection [12, 23, 27]. Recently, researchers fine-tuned

(a) Med-LMM

(b) Generate MCV

(c) Apply MCV

**Figure 2: Overview of our proposed MCVGen, which contains three key steps. (a) The architecture of our proposed Med-LMM. It utilizes the MedCLIP-ViT to encode image representations and adopts the LLaMa-2 to generate medical reports. (b) The illustration for generating MCV on a forward pass on multi-modal demonstrations. $h_i$ refers to outputs of the $i$-th transformer hidden layer at the last token position. (c) Applying MCV for new queries to generate new writing-style reports or describe novel diseases.**

LLMs with cracking diagnosis-driven prompts [15] to further improve the quality of predicted reports. To conserve the extensive medical paired data required for training these models, another line of research investigating alternative learning paradigms for MRG has attracted increasing attention, due to its potential in clinical practice. These methods can be divided into two categories. On the one hand, researchers explore extra knowledge graph sharing latent space to bridge the visual and textual domains. For example, Liu *et al.* [28] pre-constructed a knowledge graph and integrated it with visual features or linguistic information to predict reports. Similarly, Jia *et al.* [14] extracted visual features and projected them into node features. Then, a Graph Convolutional Network (GCN) [18] was adopted to model and strengthen the intrinsic correlations among diseases, allowing knowledge transfer from regular diseases to those rare diseases. On the other hand, cheaper and more easily accessible additional annotations are utilized to train the models. Both Han *et al.* [8] and Sun *et al.* [34] proposed lesion-guided weakly supervised report generation methods to describe novel chest and retinal diseases. The pre-trained lesion detectors can extract lesion-centric features by detecting abnormal regions to learn correlations between based and novel classes. Adopting lesion detectors also improves the explainability of MRG systems.

Different from these works, our approach harnesses the zero-shot capabilities of LLMs to generate novel writing-style reports or describe novel diseases through our proposed MCV. The MCV can represent fully intended task information and obviate the need for additional knowledge or annotations.

## 2.2 In-Context Learning

In-context learning (ICL) has revolutionized the approach to leveraging pre-trained LLMs for a wide array of tasks without the need

for explicit retraining or fine-tuning. This paradigm shift is predicated on the model's ability to infer task-specific instructions from very few demonstrations within the same context as the query. This method exploits the inherent capability of LLMs to generate predictions based on the contextual clues provided, making it an efficient strategy for extending the utility of these models to new tasks [7]. However, the performance of ICL highly relies on the selections of templates and demonstrates, especially in the multi-modal domain [44]. Recent works explore the underlying mechanism behind ICL and propose several kinds of vectors [9, 29, 36] to alter the ICL. Due to the auto-regressive mechanism, the last hidden state can contain essential contextual information. For instance, Hendel *et al.* [9] compresses a training set into a single task vector, which then modulates the LLM to generate outputs. Furthermore, Todd *et al.* [36] reveals that LLMs possess a neural mechanism for encapsulating input-output functions as compact and distinct from task vectors, being robust across contexts and capable of semantic vector composition, thus enabling the execution of complex new tasks. Compared to these methods, our proposed MCV represents multi-modal demonstrations, generating a unique MCV for each demonstration without any template. To the best of our knowledge, our work is the first to apply this concept in the medical multi-modal domain to encode task-specific information into MCV representations.

## 3 METHOD

The detailed pipeline of our MCVGen is presented in Fig.2, it has three critical steps to harness the zero-shot capabilities of LLMs to generate new writing-style reports or describe novel diseases. First, we pre-train a Medical Large Multi-modal Model (Med-LMM) that can achieve comparable report generation performances with existing SOTA methods. Second, we use a forward pass on each

multi-modal demonstration to attain MCV from the last hidden state of the Med-LMM. Last, MCV is employed to shift the latent space of the Med-LMM on a new query.

## 3.1 Medical Large Multi-modal Model

Unlike the natural domain, which boasts numerous open-source Visual Language Models (VLMs) with exemplary multi-modal reasoning and generation capabilities, existing open-source Medical VLMs (Med-VLMs) tend to focus primarily on Medical Visual Question Answering (Med-VQA) capabilities. In response, we first pre-train a Medical Large Multi-modal Model (Med-LMM) with report generation capabilities and then validate the zero-shot capability of our proposed MCV.

Med-LMMs typically feature a straightforward and cohesive design, incorporating a vision encoder, a Visual-Prompt Generator (VPG), and a pre-trained Large Language Model (LLM) to furnish LMMs with the capability to generate multimodal content. The role of the VPG, as discussed in works [1, 20] involves translating the image features processed by the vision encoder into visual embeddings. These embeddings are then utilized to bridge the understanding between LLMs and the visual context presented. Our Med-LMM architecture, detailed in Fig.2 (a), utilizes the MedCLIP-ViT [42] model for vision encoding to convert images into representations, labeled $Img$. Subsequently, these representations are processed by the Q-former [20] to transform image data into a format that LLMs can interpret, optimizing the number of visual tokens to enhance computational efficiency and aligning the dimension of visual embeddings with that of the text embedding of the LLM. In this study, we integrate the LLaMa-2 [37] (Hugging Face id is LlaMa-2-7b-chat-hf) model as our LLM choice. A special token $[IMG]$ is introduced to the middle of prompts and visual embeddings, aiding the LLM in differentiating between visual and textual inputs. In this manner, image and text embeddings can be directly fed into the LLM and treated equally, without employing extra gated cross-attention layers and modifying the architecture of the LLM [1].

**Pre-training Paradigm.** We employ instruction tuning on the LLM to pre-train the whole Med-LMM, specifically focusing on the tokens that comprise the report, while adhering to its initial auto-regressive learning objective. Consider a report with length $L_r$, based on an instructional prompt $X_p$ and visual embeddings $X_v$. The loss function is then defined through the negative log-likelihood as follows:

$$\mathcal{L}(\theta; X_p, X_v, X_r) = - \sum_{i=1}^{L_r} \log p_\theta(x_i | X_p, X_v, X_{r,<i}), \quad (1)$$

where $\theta$ represents the parameters subject to training, and $X_{r,<i}$ denotes the sequence of report tokens preceding the current token $x_i$ to be predicted. $X_p$ is "Generate a comprehensive and precise report for this Chest X-ray image." We froze both MedCLIP-ViT and LLaMa-2 during pre-training, and all trainable parameters only come from the Q-former.

## 3.2 Multi-modal Contextual Vector

After preparing a Med-LMM that harnesses the power of the LLM to generate reports, the next step is to construct MCV from given demonstrations. The auto-regressive nature of LLMs ensures that the hidden states at any given token position encapsulate the contextual relationships among all preceding tokens. This principle forms the foundation of ICL, enabling the effective generation of MCV that are rich in contextual information derived from multi-modal data inputs. Let us represent our pre-trained Med-LMM by $f(\cdot)$, given a multimodal demonstration $X$, which includes prompts $X_p$, images $X_v$, and the corresponding report $X_r$, we generate the Multimodal Contextual Vector (MCV) as: $h_{MCV} = f([X_p, X_v, X_r])$. More precisely, an MCV encapsulates the outputs from each hidden layer of the large language model at the position of the last token. Thus, the MCV can be expressed as $h_{MCV} = \{h_1, h_2, \ldots, h_L\}$, where $L$ denotes the number of hidden layers in the LLM, $h_i \in \mathbb{R}^d$. For the purposes of this study, we employ LLaMa-7B, which comprises a total of $L = 32$ hidden layers, and $d = 4096$ is the dimension of text embeddings.

Medical reports are typically more extensive than captions for natural images, resulting in multi-modal demonstrations that can span 450-700 tokens, encompassing prompts, visual, and textual elements. Generating a single MCV for multiple demonstrations is inefficient and necessitates the use of larger backbone networks. Therefore, we generate one MCV for each demonstration and compute the average output of each layer to obtain the final task-specific MCV. This MCV offers three primary advantages. Firstly, it is more efficient and conserves computational resources, enabling the use of a greater number of demonstrations. Secondly, it prevents the model from overly relying on the contextual information from the most recent demonstrations when generating new reports, ensuring a more balanced and comprehensive understanding of the task at hand. Finally, we employ the outputs from each hidden layer, opting to bypass the selection of activations within the multi-head self-attention modules [36]. This approach is straightforward and readily adaptable to a variety of tasks.

## 3.3 MCV Guides New Query

As discussed by Liu *et al.*, the rationale for utilizing vector representations to alter ICL can be formulated as:

$$Att(x_{query}W_q, XW_k, XW_v) =: \alpha h(X_{query}) + (1 - \alpha)h(X_{demo}), \quad (2)$$

where $Att(\cdot)$ refers to the multi-head self-attention module in each layer, $X = [X_{demo}; X_{query}]$ denotes the inputs and $x_{query}$ is the token in $X_{query}$. Then, $h(X_{demo})$ can replaced by our proposed MCV. Therefore, the overview process of employing MCV to control the new query generation can be written as follows:

$$Att(x_{query}W_q, XW_k, XW_v) =: \alpha h(X_{query}) + (1 - \alpha)h_{MCV}, \quad (3)$$

More specifically, after securing the MCV, we execute a forward pass on the query example and add the MCV to the hidden states at the first query token position at all layers $l = 1, 2, \ldots, L$ as:

$$\tilde{h}_l = h_l + h_{MCV}^l, \quad (4)$$

To retain the model's inherent capabilities to the fullest extent, we normalize the adjusted hidden states to align with the $\ell_2$ norm of the hidden states prior to modification:

$$\tilde{h}_l = \tilde{h}_l \cdot \frac{\|h_l\|_2}{\|\tilde{h}_l\|_2}.$$

| Method | IU-Xray | | | | Method | MIMIC-CXR | | | |
|---|---|---|---|---|---|---|---|---|---|
| | BLEU-4 | ROUGE | METEOR | CIDEr | | BLEU-4 | ROUGE | METEOR | CIDEr |
| R2Gen | 0.165 | 0.371 | 0.187 | - | R2Gen | 0.103 | 0.277 | 0.142 | - |
| PPKED | 0.168 | 0.376 | 0.190 | 0.351 | PPKED | 0.106 | 0.284 | 0.149 | 0.237 |
| DCL | 0.163 | **0.383** | 0.193 | **0.586** | DCL | 0.109 | 0.284 | 0.150 | **0.281** |
| MET | 0.172 | 0.380 | 0.192 | 0.435 | PromptMRG | 0.112 | 0.268 | 0.157 | - |
| R2GenGPT-S | 0.156 | 0.370 | 0.202 | 0.405 | R2GenGPT-S | 0.117 | 0.277 | 0.136 | 0.145 |
| R2GenGPT-D | **0.173** | 0.377 | **0.211** | 0.438 | R2GenGPT-D | **0.134** | **0.297** | 0.160 | 0.269 |
| Med-LMM (**ours**) | 0.168 | 0.381 | 0.209 | 0.427 | Med-LMM (**ours**) | 0.128 | 0.289 | **0.161** | 0.265 |

**Table 1: Comparison of fully supervised MRG performance across NLG metrics on IU-Xray and MIMIC datasets. The results are quoted from their original papers. The highest performances for each metric are highlighted in bold.**

| Method | Precision | Recall | F1-score |
|---|---|---|---|
| R2Gen | 0.333 | 0.273 | 0.276 |
| DCL | 0.471 | 0.352 | 0.373 |
| PromptMRG | **0.501** | **0.509** | **0.476** |
| R2GenGPT-S | 0.341 | 0.312 | 0.325 |
| R2GenGPT-D | 0.392 | 0.387 | 0.389 |
| Med-LMM (**ours**) | 0.412 | 0.373 | 0.395 |

**Table 2: Comparison of fully supervised MRG performance across CE metrics on MIMIC dataset. The highest performances for each metric are highlighted in bold.**

This normalization ensures that the adjusted latent state vectors maintain a magnitude compatible with the expectations of the model's subsequent processing stages.

## 4 EXPERIMENT

### 4.1 Implementation Details

**Datasets.** Our work utilizes two MRG datasets available to the public for both pre-training the Med-LMM and assessing the zero-shot learning capability of proposed MCVGen. The *IU-Xray* dataset [6] is a common benchmark for testing radiology report generation systems, featuring 3,955 radiological reports paired with 7,470 chest X-ray photographs. These images may present in frontal alone or both frontal and lateral views per report. The *MIMIC-CXR* database [17] stands as the most extensive collection in this field, encompassing 368,960 chest X-ray visuals alongside 222,758 corresponding reports, with an official division for dataset splitting. Additionally, each image-report pair is annotated with multiple labels for 14 diseases using the CheXPert [13] labeling tool. The MIMIC dataset is also compiled as a corpus for pre-training a range of medical VLMs. Following the pre-processing methodology outlined in R2Gen [5], we exclude all studies lacking either reports or images. Consequently, the final train/validation/test splits for the IU-Xray and MIMIC datasets are 2069/296/590 and 270790/2130/3858 pairs, respectively.
**Metrics.** We utilize two kinds of metrics to automatically evaluate the quality of predicted reports. *Natural Language Generation Metrics* (NLG) serve as benchmarks for assessing the descriptive precision of generated reports. BLEU-4 [32] and CIDEr [39] are the primary metrics adopted in existing MRG works for comparison. Originally designed for assessing machine translation systems,

BLEU calculates the similarity between the generated text and reference reports based on the overlap of word n-grams. CIDEr, on the other hand, is more suited for captioning tasks as it emphasizes the significance of topic-specific terminology (critical in MRG tasks) while downplaying common phrases. For a broader evaluation, we also include results from ROUGE-L [24] and METEOR [2] metrics. *Clinical Efficacy Metrics*, a newer set of benchmarks, focus on the clinical correctness of the generated reports. This involves using the CheXPert labeling tool to annotate predicted reports with 14 diseases, followed by the computation of classification metrics such as F1-Score, Precision, and Recall to gauge the reports' ability to accurately depict medical conditions. Due to the original IU-Xray dataset not being annotated with CheXPert, we also evaluate the clinical efficacy metrics on the MIMIC dataset. More details about demonstration selection are in the Appendix.
**Training Details.** In this work, we leverage the MedCLIP-ViT to encode image representations, Q-former to extract visual embeddings and LLaMa-2 to generate reports. When pre-training the Med-LMM, we froze the MedCLIP-ViT and LLama-2. During the pre-training, the trainable parameter is 192M, including the Q-former with 32 learnable queries whose dimension is 768 and a linear layer to project the dimension to 4096 fitting the dimension of LLaMa-2. We used mix-precision and pre-trained the Med-LMM for 5 and 15 epochs on the IU-Xray and MIMIC, respectively. The learning rate is set as $1e-4$ and the optimizer is Adam with a weight decay of 0.02. It requires 0.18 seconds to train a batch with a size is 8. All the experiments were conducted on 4 NVIDIA RTX A5000 24GB GPUs, and implemented by Python 3.11.

### 4.2 Zero-shot Settings

In this section, we detail the approach of leveraging the IU-Xray and MIMIC datasets to assess the zero-shot capabilities of MCVGen. We simulate two clinical scenarios: cross-center evaluation and cross-disease evaluation.

For the **cross-center valuation**, our approach stems from a practical consideration: smaller medical centers often lack the resources to gather a sufficient volume of paired data needed to support the fine-tuning of Med-LMMs. Additionally, there are significant disparities in report writing styles and terminology use across different centers. This variability presents a challenge for model generalization and underscores the importance of evaluating model performance across diverse clinical settings. Specifically,

| Method | IU-Xray->MIMIC-CXR | | | | Method | MIMIC-CXR->IU-Xray | | | |
|---|---|---|---|---|---|---|---|---|---|
| | BLEU-4 | ROUGE | METEOR | CIDEr | | BLEU-4 | ROUGE | METEOR | CIDEr |
| R2Gen | 0.067 | 0.201 | 0.108 | 0.072 | R2Gen* | 0.059 | 0.253 | 0.131 | - |
| R2GenCMN | 0.061 | 0.213 | 0.105 | 0.069 | DCL* | 0.074 | 0.267 | 0.152 | - |
| DCL | 0.064 | 0.208 | 0.105 | 0.063 | PromptMRG* | 0.098 | 0.281 | 0.160 | - |
| R2GenGPT-S | 0.064 | 0.211 | 0.108 | 0.066 | R2GenGPT-S | 0.051 | 0.229 | 0.114 | 0.196 |
| + MCV | 0.088 | 0.247 | 0.118 | 0.083 | + MCV | 0.083 | 0.291 | 0.149 | 0.207 |
| R2GenGPT-D | 0.071 | 0.218 | 0.113 | 0.072 | R2GenGPT-D | 0.083 | 0.275 | 0.163 | 0.231 |
| + MCV | 0.101 | 0.251 | 0.122 | 0.134 | + MCV | 0.112 | 0.308 | 0.177 | 0.236 |
| Med-LMM | 0.068 | 0.215 | 0.115 | 0.069 | Med-LMM | 0.087 | 0.279 | 0.137 | 0.252 |
| + MCV (**MCVGen**) | 0.106 | 0.256 | 0.128 | 0.142 | + MCV (**MCVGen**) | 0.125 | 0.313 | 0.185 | 0.280 |

**Table 3: Comparison of cross-center zero-shot learning performance across NLG metrics on the IU-Xray and MIMIC dataset. The datasets on the left and right of -> indicate the pre-training and testing sets, respectively. * indicates the results are quoted from their original papers.**

| Method | Precision | Recall | F1-score |
|---|---|---|---|
| R2Gen | 0.139 | 0.134 | 0.135 |
| R2GenCMN | 0.145 | 0.132 | 0.134 |
| DCL | 0.156 | 0.143 | 0.146 |
| R2GenGPT-S | 0.128 | 0.116 | 0.119 |
| + MCV | 0.181 | 0.169 | 0.173 |
| R2GenGPT-D | 0.208 | 0.193 | 0.197 |
| + MCV | 0.242 | 0.226 | 0.234 |
| Med-LMM | 0.211 | 0.206 | 0.208 |
| + MCV (**MCVGen**) | 0.249 | 0.235 | 0.237 |

**Table 4: Comparison of cross-center zero-shot learning performance across CE metrics in the IU-Xray to MIMIC-CXR adaptation.**

we applied the proposed MCV to guide a Med-LMM, initially pre-trained on the IU-Xray dataset, in describing MIMIC images, and vice versa.

Regarding the **cross-disease evaluation**, our simulation addresses a scenario where a new center faces a novel disease. Due to the rarity or emergent nature of the disease, the center may not be able to collect a sufficient number of samples to train or fine-tune models effectively. This scenario tests the model's ability to generalize to new diseases without the benefit of extensive, disease-specific data, reflecting the real-world challenge of adapting to emerging health threats with limited information. The cross-disease evaluation was exclusively conducted on the MIMIC dataset. We adopted the leave-one-out approach, where in each iteration, data from one disease on the test subset is selected for testing, while the data without this disease on the training subset are used to pre-train the model. The hyper-parameters used in this pre-training remain consistent with those specified earlier.

## 4.3 Fully Supervised Report Generation Performances

We validated the fully supervised MRG performances of the Med-LMM on two datasets. It can verify the effectiveness of leveraging

LLM to generate medical reports and can further help us understand the ceiling performances of MVCGen. We compared our MCVGen with several SOTA MRG systems, such as R2Gen [5], R2GenCMN [4], PPKED [27], DCL [22], MET [40], R2GenGPT [41] and PromptMRG [15].

Notably, the primary goal of introducing Med-LMM is not to claim STOA results in MRG but to present a medical vision large model with the potential for zero-shot learning. Hence, we have not strictly replicated existing works; instead, we provide a general comparison with the results presented in their original publications, showcasing the capabilities of our Med-LMM. Tab.1 provides a general comparison of fully supervised MRG performance across NLG metrics on the IU-Xray and MIMIC datasets. Our proposed Med-LMM exhibits competitive performance, almost matching the best-reported BLEU-4 score of 0.128 vs 0.134 on the MIMIC dataset, a notable achievement. While Med-LMM's performance on the IU-Xray dataset does not reach the peak in any single metric, it shows a robust balance across all considered NLG metrics, with scores closely trailing the best in METEOR (0.209 vs. 0.211) and nearly matching the top performance in ROUGE (0.381 vs. 0.383). On the MIMIC dataset, Med-LMM ties for the highest METEOR score and demonstrates commendable performance in ROUGE (0.289) and BLEU-4 (0.128), underscoring its effectiveness in generating descriptively correct reports. In addition, Tab.2 presents a comparison of fully supervised MRG performance across CE metrics within the context of the MIMIC dataset. Notably, the PromptMRG outperforms other approaches, securing the highest metrics across the board. Our proposed Med-LMM exhibits a balanced performance with a Precision of 0.412, Recall of 0.373, and a macro F1-score of 0.295. Such results reflect its capability to accurately generate clinically relevant content.

The observation across the two tables underscores an intriguing point about the performance of our Med-LMM model. It holds its ground against the R2GenGPT-D (Deep) and significantly surpasses the R2GenGPT-S (Shallow). The R2GenGPT framework integrates the Swin Transformer [30] as its vision encoder, a simple linear layer as the mapper, and LLaMa-2 for decoding. In its shallow configuration, both the Swin Transformer and LLaMa-2 are frozen, with only the mapper's 4.2 million parameters being trainable. Conversely,

| Method | NLG Metric | | | | CE Metric | | |
|---|---|---|---|---|---|---|---|
| | BLEU-4 | METEOR | ROUGE | CIDEr | Precision | Recall | F1-score |
| R2GenGPT-S | 0.098 | 0.253 | 0.122 | 0.093 | 0.173 | 0.161 | 0.163 |
| + MCV | 0.105 | 0.267 | 0.132 | 0.121 | 0.192 | 0.179 | 0.180 |
| R2GenGPT-D | 0.117 | 0.277 | 0.136 | 0.144 | 0.253 | 0.238 | 0.244 |
| + MCV | 0.124 | 0.285 | 0.145 | 0.187 | 0.289 | 0.267 | 0.275 |
| Med-LMM | 0.113 | 0.269 | 0.141 | 0.140 | 0.261 | 0.232 | 0.246 |
| + MCV (**MCVGen**) | 0.129 | 0.291 | 0.148 | 0.203 | 0.294 | 0.277 | 0.281 |

**Table 5: Comparison of cross-disease zero-shot learning performance across NLG and CE metrics on MIMIC dataset.**

(a) Average MCV (ours)    (b) Recurrent MCV

(c) Shift first position (ours)  (d) Shift all positions  (e) Replace first position

**Figure 3: Illustrations of different settings to generate and apply MCV. (a) Our MCV by averaging all vectors. (b) Recurrently shift the first position in the next demo and use the last output. (c) Our MCV to shift the first position in the query. (d) Shift all positions in the query. (e) Replace the hidden states in the first position by MCV.**

| Setting | NLG Metric | | CE Metric | | |
|---|---|---|---|---|---|
| | BLEU-4 | CIDEr | Precision | Recall | F1-Score |
| a+c | 0.106 | 0.142 | 0.249 | 0.235 | 0.237 |
| b+c | 0.073 | 0.077 | 0.228 | 0.211 | 0.215 |
| a+d | 0.102 | 0.138 | 0.245 | 0.229 | 0.238 |
| a+e | 0.098 | 0.114 | 0.206 | 0.189 | 0.192 |

**Table 6: Comparison of different settings to generate and apply MCV in the IU-Xray to MIMIC adaptation. The setting "a+c" represents our MCVGen.**

the deep model allows full training of both the Swin Transformer and the mapper, engaging a total of 90.9 million parameters. This difference in approach between the shallow and deep models of R2GenGPT highlights the significance of training depth in achieving higher performance metrics. Our Med-LMM model, leveraging the MedCLIP-ViT as its foundation with a trainable Q-former, demonstrates that it doesn't require additional training on the data of downstream tasks to achieve competitive results.

## 4.4 Zero-shot Learning Results

**Cross-center Performance.** Tab.3 and Tab.4 showcase a detailed comparison of cross-center zero-shot learning performance across NLG and CE metrics, respectively. Notably, the incorporation of the proposed MCV, denoted as "+ MCV" in the table, significantly enhances the performance across all metrics for baseline models engaged in this study. For instance, in the IU-Xray to MIMIC-CXR adaptation, the R2GenGPT-D model improved from a BLEU-4 score of 0.071 to 0.101, ROUGE from 0.218 to 0.251, METEOR from 0.113 to 0.122, and CIDEr from 0.072 to 0.134 after incorporating MCV. This indicates a substantial enhancement, particularly in the CIDEr metric, which nearly doubled. Similarly, for the MIMIC-CXR to IU-Xray direction, the same model with MCV added saw improvements in BLEU-4 from 0.083 to 0.112, ROUGE from 0.275 to 0.308, METEOR from 0.163 to 0.177, and CIDEr from 0.231 to 0.236. The most striking advancement is observed with our model, MCVGen (Med-LMM + MCV), which achieves the highest performance improvements in both directions. Specifically, for IU-Xray to MIMIC-CXR, MCVGen boosts the BLEU-4 score from 0.068 to 0.106, ROUGE from 0.215 to 0.256, METEOR from 0.115 to 0.128, and CIDEr from 0.069 to 0.142. For the reverse direction, the enhancements are even more pronounced, with BLEU-4 jumping from 0.087 to 0.125. The enhancements across NLG metrics illustrate that incorporating the MCV can foster the generation of reports with a new writing style, due to greater word overlap that stems from similar writing styles. Comparable improvements are observed across CE metrics in the IU-Xray to MIMIC-CXR adaptation. By integrating MCV with R2GenGPT-S, R2GenGPT-D, and Med-LMM, the precision is increased by 5.3%, 3.4%, and 3.8%, respectively. These enhancements highlight that the generated reports can capture accurate medical terminologies, thereby enhancing clinical factual correctness.

Overall, these results demonstrate that our MCVGen achieves SOTA zero-shot MRG performances on both datasets and even surpasses some fully supervised MRG systems. The impact of incorporating MCV underscores its ability to leverage the unique strengths of LLMs. When comparing MCVGen and R2GenGPT-D+MCV, the advantages of utilizing Q-former, which provides stronger visual embedding capabilities, become evident.

**Cross-disease Performance.** We present the cross-disease evaluation results across both NLG and CE metrics on the MIMIC in Tab.5. It is observed that incorporating MCV can enhance the baseline models on both NLG and CE metrics. In terms of NLG metrics, integrating MCV with R2GenGPT-S improved BLEU-4 scores from 0.098 to 0.105. For R2GenGPT-D, the introduction of MCV resulted in

| | Clinical Information | Ground Truth | Med-LMM | MCVGen |
|---|---|---|---|---|
| | Study Id: 59358522
Subject Id: 12110863

CheXpert Label:
Cardiomegaly
Airspace Opacity
Fracture | impression: Right lower lobe fibrosis. Moderate cardiomegaly. Interval right humeral neck fracture. Findings: Peripheral fibrosis and mild architectural distortion in the right lower lobe. No focal consolidation. Pulmonary edema has resolved. Bilateral pleural thickening. Right atrial and ventricular pacemaker leads, the latter coursing in the mid RV. Median sternotomy wires and mediastinal clips. Moderate-to-severe cardiomegaly is unchanged. Aorta is tortuous and unfolded. Multilevel degenerative changes in the thoracic spine. Interval fracture of the right humeral surgical neck, with an overriding fracture fragment. This appears subacute, with partially corticated margins. | impression : no evidence of pneumothorax. findings : the right upper lobe of the left lung has been removed for imaging. there is no evidence of pleural effusion or pneumothorax. the right lung is clear. there is no evidence of acute pulmonary edema. the cardiac silhouette appears normal in size and contour. | impression: no evidence of pneumothorax. interval changes with right lower lobe fibrosis and moderate cardiomegaly. findings: in comparison with the study of the monitoring and support devices remain in place. lateral view somewhat limited due to overlying motion artifact. no pleural effusions or pulmonary edema. there is no pneumothorax. the inferior sternotomy wire is fractured but unchanged. |

**Figure 4: Qualitative examples from the cross-disease evaluation, where we present the clinical information, ground truth, and predictions from Med-LMM and MCVGen. Text highlighting novel disease categories is shown in red.**

in an increase in BLEU-4 scores from 0.117 to 0.124 and CIDEr from 0.144 to 0.187. Our MCVGen demonstrated the most pronounced gains, with BLEU-4 scores rising from 0.113 to 0.129 and CIDEr from 0.140 to 0.203. On the CE front, the incorporation of MCV into R2GenGPT-S and R2GenGPT-D boosted the precision from 0.173 to 0.192 and 0.253 to 0.289, respectively. MCVGen also achieved significant improvements in precision from 0.261 to 0.294, recall from 0.232 to 0.277, and F1-score from 0.246 to 0.281. These results clearly illustrate the significant impact that MCV has on enhancing the performance of MRG systems in cross-disease zero-shot learning contexts. The enhancements in CE metrics that are more important in the cross-disease evaluation highlight MCV's role in increasing the clinical factual correctness of the reports, crucial for practical medical applications. It shows the potential of deploying MCVGen in clinical practice to describe novel diseases with a few demonstrations.

### 4.5 Ablation Study for MCV

In this section, we conduct ablation studies to compare different settings to generate and apply MCV. The illustrations of different settings are shown in Fig.3. The upper figures show the ways how we generate MCV. In setting (b), we proposed a recurrent MCV for comparison. In which, we applied the MCV from the next demo to shift the next demo's first hidden states. And applied the MCV generated from the last demo to the new query. In setting (d), we applied MCV to shift hidden states in all query token positions, similar with the use from ICV [29]. For setting (e), we replace the first hidden states by MCV, following the approach by Task Vectors (TV) [9]. Setting (b+c) introduces a recurrent MCV approach, resulted in lower performance compared to setting (a+c), with BLEU-4 at 0.073, CIDEr at 0.077, and CE metrics at 0.228, 0.211, and 0.215, respectively. We also notice that the output highly similar to the last demo report. It is necessary to get the average of all demo MCVs and then guide the query generation process. The performances between setting a+c and a+d are comparable. We speculate the reason for these slight drops is that incorporating MCV into every position attracts the attention from query itself, despite applying the $\ell_2$ normalization. In the end, setting a+e follows an approach similar to the TV and yielded lower NLG and CE metrics than the MCVGen, with BLEU-4 at 0.098 vs 0.106, CIDEr at 0.114 vs 0.142, and precision, recall, and F1-score at 0.206, 0.189, and 0.192, respectively. It proves the importance to keep the first original token.

These ablation studies underscore the effectiveness of our MCV-Gen approach (a+c) in harnessing the full potential of MCV for enhancing zero-shot MRG.

### 4.6 Case Study

We conduct a case study and present the quality example in Fig.4 to depict the exceptional cross-disease report generation capabilities of our proposed MCVGen. During this iteration, the "*facture*" was selected as the novel disease and demonstration. Comparing the predicted reports between Med-LMM and MCVGen, we can observe that incorporating the MCV containing novel contextual information can rephrase the generation process and let the model describe unlearned diseases. Also, MCVGen can predict a true positive for "cardiomegaly". The ability to generalize across varied clinical settings and disease types underscores MCVGen's potential to alleviate the burden on radiologists by aiding in the generation of accurate and efficient medical reports. This could be particularly advantageous in under-resourced environments where the scarcity of qualified radiologists delays diagnoses and treatments.

## 5 CONCLUSION

In this paper, we propose MCVGen, a novel approach for zero-shot medical report generation that harnesses the exceptional capabilities of large language models by incorporating the multi-modal contextual vector (MCV) to alter traditional multi-modal in-context learning. Initially, we pre-trained a Med-LMM, and then constructed the MCV by averaging the last hidden states from each demonstration on forward passes. We introduced two zero-shot settings and conducted experiments on the IU-Xray and MIMIC datasets. The results demonstrate that our MCV effectively controls the Med-LMM to generate reports in new writing styles or about novel diseases, achieving state-of-the-art zero-shot MRG performances across both natural language generation and clinical efficacy metrics. It demonstrates the potential of deploying MCVGen in clinical practice to accelerate the development of healthcare AI.

**Limitations.** The promising results of our approach largely depend on leveraging powerful LLMs. Due to limited GPU resources, LLaMa-2 7B is the largest LLM we could utilize. We are exploring efficient ways to harness more powerful LLMs. Moreover, our current experiments focus solely on the chest radiology modality. Future work will assess the cross-modality zero-shot learning capabilities of MCVGen.

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
