# OpenReview forum: "In-Context Learning for Zero-shot Medical Report Generation"
_acmmm.org/ACMMM/2024/Conference — MM2024 Poster_

### Official Review · Reviewer_Qm2r · 2024-05-21

**Rating:** 4
**Confidence:** 3

**Summary:**

This paper introduces a Multi-Modal Contextual Vector (MCV) to enhance multi-modal in-context learning for zero-shot MRG, and its effectiveness is demonstrated through relevant experiments. However, there are some issues with clarity that need to be addressed.

**Strengths:**

The idea of using a Multi-Modal Contextual Vector (MCV) to enhance the zero-shot capability of large language models (LLMs) for medical report generation is quite interesting. Furthermore, the ablation studies explore various methods of utilizing MCV, which adds to the reliability of the results.

**Limitations:**

1. In Section 3.2, what is the definition and source of the demonstrations? Does this refer to information from similar/relevant/retrieved samples? In other words, what do demo1, demo2, and demo3 in Figure 2 specifically mean? These details need to be clarified in the paper.

2. The authors claim that the proposed model is efficient, controllable, and robust, but these characteristics have not been validated through relevant experiments, particularly in terms of controllability and robustness.

3. The paper contains numerous detailed issues. For example: "Open IU" mentioned in line 94; missing citation on line 441; an extra parenthesis after Equation 3; and inconsistent font sizes in Figure 4.

4. More qualitative experiments are supported be added to validate the differences in report generation under various model settings.

**Suitability:**

3

---

### Official Review · Reviewer_agsh · 2024-05-23

**Rating:** 5
**Confidence:** 3

**Summary:**

To address the challenge of collecting large amounts of paired medical image-report data for trainin, this paper proposes a zero-shot report generation model based on in-context learning, called MCVGen. It is a novel approach that harnesses the exceptional capabilities of large language models by incorporating the multi-modal contextual vector (MCV) to alter traditional multi-modal in-context learning. This paper introduced two zero-shot settings and conducted experiments on the IU-Xray and MIMIC datasets, demonstrating the potential of deploying MCVGen in clinical practice.

**Strengths:**

1. The innovation is commendable and it is the first work to leverage LLMs for zero-shot medical report generation.
2.Based on the experimental results, the model design is meaningful. The introduction of MCV significantly enhances the model's report generation capability.

**Limitations:**

1.In line 859, "We also notice that the output is highly similar to the last demo report." The authors are encouraged to provide a visual representation.
2.The authors are encouraged to explain from a model design perspective why MCVGen can identify new diseases.
3. In line 94, the experimental section states that the model is tested on the IU-Xray and MIMIC datasets, which is inconsistent with the title of the abstract.
4. In line 452, Equation (3) has an extra parenthesis.
5. It is recommended to bold the best experimental data in the tables for easier comparison.
6. Section 4.5 should mention Table 6.

**Suitability:**

3

---

### Official Review · Reviewer_6uVx · 2024-05-23

**Rating:** 5
**Confidence:** 4

**Summary:**

The paper proposes a zero-shot medical report generation model based on in-context learning. It introduces a multi-modal contextual vector (MCV) to represent the contextual information of demonstrations and uses it to guide the generation of new reports. The model is tested on two publicly available datasets, demonstrating exceptional zero-shot capabilities on cross-center and cross-disease evaluations.

**Strengths:**

- The paper addresses the challenge of collecting large amounts of paired medical image-report data for training by proposing a zero-shot learning approach.
- The use of in-context learning and the introduction of MCV provide a novel and efficient way to generate medical reports.
- The model achieves state-of-the-art zero-shot performance on both cross-center and cross-disease evaluations, demonstrating its effectiveness.

**Limitations:**

- The paper does not provide a detailed comparison with existing methods in terms of performance metrics.
- The evaluation is limited to two datasets, and it would be beneficial to see the performance of the model on other datasets as well.
- The paper does not discuss potential limitations or challenges of the proposed approach.

**Suitability:**

3

---

### Meta-Review · Area_Chair_VAUd · 2024-06-23

**Recommendation:** Accept (Poster)
**Confidence:** 5

**Metareview:**

The paper proposes a novel zero-shot medical report generation model based on in-context learning, termed MCVGen. The model introduces a multi-modal contextual vector (MCV) to represent the contextual information of demonstrations, enhancing the generation of new reports. The approach innovates by departing from traditional in-context learning methods, leveraging the last hidden state of each demonstration to guide the generation process. This reduces computational costs while maintaining high accuracy. The model is tested on the Open-IU and MIMIC datasets, demonstrating exceptional zero-shot capabilities in cross-center and cross-disease evaluations.

Pros:
+ The paper is well-structured and written.
+ The use of MCVs in conjunction with in-context learning is a novel contribution to the field of medical report generation.
+ The model achieves state-of-the-art zero-shot performance on both cross-center and cross-disease evaluations, demonstrating its effectiveness and robustness.
+ The paper includes detailed experiments on the Open-IU and MIMIC datasets, showcasing the model's capabilities and potential for practical deployment.

Cons:
+ The evaluation is limited to two datasets. Additional experiments on other datasets would strengthen the validity of the results.
+ The paper lacks a detailed comparison with existing methods.

Given the three positive reviews, I recommend accepting this paper.